# Biodegradation Potential of C_7_-C_10_ Perfluorocarboxylic Acids and Data from the Genome of a New Strain of *Pseudomonas mosselii* 5(3)

**DOI:** 10.3390/toxics11121001

**Published:** 2023-12-08

**Authors:** Sergey Chetverikov, Gaisar Hkudaygulov, Danil Sharipov, Sergey Starikov, Darya Chetverikova

**Affiliations:** Ufa Institute of Biology, Subdivision of the Ufa Federal Research Centre of the Russian Academy of Sciences, 450054 Ufa, Russia; che-kov@mail.ru (S.C.); male886@yandex.ru (D.S.); senik0406@gmail.com (S.S.); belka-strelka8031@yandex.ru (D.C.)

**Keywords:** perfluorocarboxylic acids, biodegradation, transformation, *Pseudomonas*, genome

## Abstract

The use of bacteria of the genus *Pseudomonas*—destructors of persistent pollutants for biotechnologies of environmental purification—is an interesting area of research. The aim of this work was to study the potential of *Pseudomonas mosselii* strain 5(3) isolated from pesticide-contaminated soil as a degrader of C_7_-C_10_ perfluorocarboxylic acids (PFCAs) and analyze its complete genome. The genome of the strain has been fully sequenced. It consists of a chromosome with a length of 5,676,241 b.p. and containing a total of 5134 genes, in particular, haloalkane dehalogenase gene (*dhaA*), haloacetate dehalogenase H-1 gene (*dehH1*), fluoride ion transporter gene (*crcB*) and alkanesulfonate monooxygenase gene (*ssuE*), responsible for the degradation of fluorinated compounds. The strain *P. mosselii* 5(3) for was cultivated for 7 days in a liquid medium with various C_7_-C_10_ PFCAs as the sole source of carbon and energy, and completely disposed of them. The results of LC-MS analysis showed that the transformation takes place due to perfluorohexanoic acid with the release of various levels of stoichiometry (depending on PFCA) of fluorine ion mineralization indicators determined by ion chromatography. Thus, *Pseudomonas mosselii* strain 5(3) demonstrates a genetically confirmed high potential for the decomposition of C_7_-C_10_ PFCA.

## 1. Introduction

Perfluorocarboxylic acids (PFCAs) have recently been subjected to close regulation by international supervisory authorities due to their lability; bioaccumulation ability; and resistance, including biostability that is based on the strength of the carbon–fluorine interaction, which forms the strongest bond ever discovered [1]. The identification of these properties characterizing their toxicity to wildlife led earlier to their inclusion in Annex B of the Stockholm Convention on Persistent Organic Pollutants [2].

The main significant representatives of this class of compounds are perfluorooctanoic acid (PFOA) and perfluorooctanesulfonate (PFOS), these synthetic chemical compounds exhibit remarkable stability and are extensively employed in the fabrication of fluoropolymers for non-stick surfaces, pesticides, protective agents for fabrics, food packaging, electroplating and as components of fire-fighting foams, such as surfactants [3,4,5,6,7,8]. As a result, PFCAs are omnipresent in the global ecosystem, migrating through soil and water systems [9,10]. Its toxicological effects on humans are characterized by various types of cancer, disorders of the immune system and liver, and influence on growth, fetal development and reproductive function [11,12,13,14].

Hence, the imperative lies in expeditiously devising effective techniques for the decomposition of PFCA compounds. Various approaches have been explored for the degradation of PFOS, including chemical treatment, high-temperature incineration, and ultrasound techniques. However, these methods have proven to be both costly and inefficient in achieving satisfactory results [15,16,17,18]. The biodegradation of PFCAs is further complicated by their unique chemical properties, specifically their intricate molecular structures and substantial molecular sizes which pose a myriad of physiological challenges for microorganisms, the absorption of which by bacteria requires specialized active transport pathways. Previously, it was assumed that microorganisms did not have time to evolve in order to decompose PFCA (especially in comparison with the favorable results of the dehalogenation of polychlorinated molecules [19]) by developing the enzymatic systems acting on them. This was due to the lack of analogues in nature that would stimulate the evolution of the necessary catabolic enzymes [20]. But recently it has been shown that the deformation of PFCA is still possible. And it occurs mainly when microbial enzymes attack fluoride-free carbon covalent bonds (for example, CS and CO), as shown during the biodegradation of PFOA and PFOS, fluorotelomers in aerobic [21,22] and anaerobic conditions [23,24,25]. Some other works show the destruction of PFOS and/or PFOA, which is not confirmed by the release of fluorine ions, suggesting that minor polyfluorinated compounds are preferably formed as a result of these biotransformations [26,27,28,29]. The successful disruption of CF bonds has been widely acknowledged as a pivotal factor in the natural breakdown of organofluorine substances, such as PFCA [30]. The elimination of fluoride effectively nullifies the discernible impact of this heteroatom on the chemical stability of the organic compound, which is attributable to its exceptional electronegativity. This not only enhances the probability of the facile decomposition of the resultant metabolites but also accentuates their overall susceptibility to breakdown [31].

In most of these studies, the destructors of organofluorine compounds are *Pseudomonads*. These microorganisms have garnered extensive research attention owing to their remarkable metabolic adaptability; they are destructors of various xenobiotics. Harnessing their unparalleled metabolic versatility, these microorganisms thrive by exclusively utilizing them as the primary carbon and energy source. The genomes of various *Pseudomonas* strains across distinct species has been meticulously analyzed and documented, for example, *P. protegens* (strain Pf-5), *P. fluorescens* (strains Pf-01, SBW25), *P. azelaica* (strains HBP1, Aramco J), *P. putida* (strain L48), *P. mendocina* (strain YMP), etc. The approximate size of *Pseudomonas* genomes is around 6 Mb [32]. A multitude of genes participate in the breakdown of diverse carbon sources and their adaptation to thrive within a specific ecological habitat.

But from the point of view of genomic data, studies on PFCA degradation genes are an open book that requires filling. Currently, a scarcity of knowledge regarding bacterial strains harboring genetic sequences responsible for encoding a diverse array of enzymes that break down PFCAs can be detected and described. The exceptions are *P. parafulva* YAB-1 and *Delftia acidovorans* D4B strains (the genus *Delftia* is close to the genus *Pseudomonas*)—exploring the potential of bacterial strains housing recognized putative halogen acid dehalogenases and fluoroacetate dehalogenase as catalysts in the breakdown of fluorinated compounds [33,34]. And the list is not so extensive; only a few strains of bacteria are known to be capable of transforming perfluorocarboxylic acids. And they are only associated with the biodegradation of PFOA and PFOS, with no mentions found in the literature regarding other PFCA compounds. The aforementioned strain *P. parafulva* YAB-1 can degrade 32.2% of perfluorooctanoic acid at an initial concentration of 500 mg/L, while *P. aeruginosa* HJ4 can break down PFOS by 67% at a concentration not exceeding 2 mg/L [26].

PFCA biodegradation researchers focus mainly on C_8_ compounds, but their bioaccumulation potential increases with increasing carbon chain length [35]. Recent studies have highlighted that short-chain PFCAs exhibit enhanced mobility, increased water solubility, and heightened resistance to degradation when compared to their long-chain counterparts [36,37].

Therefore, when assessing biodegradation, it is necessary not to focus only on PFOA and PFOS—the Stockholm Convention’s inventory of persistent fluorinated organic pollutants will likely expand this in the future—as the recently uncovered toxic attributes of short-chain PFCA have emerged. Additionally, if the adaptation of microorganisms to PFOS and PFOA has already been presented in publications, then the resistance of microorganisms to intermediate products of their defluorination and decomposition and released fluorine ions, on the one hand, and their high-molecular precursors, on the other hand, have yet to be evaluated.

In this study, a new strain called *Pseudomonas mosselii* 5(3) was tested and isolated from soil contaminated with herbicides, including fluorinated ones. This strain was identified using methods of molecular biology, as well as the analysis of morphological, physiological and biochemical characteristics. The genome sequencing, the study of genetic features, and biodegradation potential of the strain in relation to C_7_-C_10_ PFCAs were carried out. These results can be effectively applied in biotechnologies of environmental rehabilitation from contamination with organofluorine compounds.

## 2. Materials and Methods

### 2.1. Bacterial Strain

Strain 5(3) was obtained from arable soil which had been affected by pesticide contamination in the Yanaulsky district of the Republic of Bashkortostan, Russian Federation (56.11381° North latitude, 54.766226° East longitude). The soil for isolating microorganisms belonged to dark-grey forest soil (Haplic Greyzems) and was characterized by the following parameters: total humus, 6.8%; N_total_, 0.60%; P_2_O_5_, 65.4 mg/kg; K_2_O, 112.5 mg/kg; available copper and exchangeable zinc, 8.4 and 0.9 mg/kg soil, respectively; and pH of water, 5.2. The pesticide contamination consisted of a mixture including phosphorus-containing glyphosate (active ingredient—N-(phosphonomethyl)glycine), chlorine-containing octapone (active ingredient—2-(2,4-dichlorophenoxy)acetic acid), and fluorine-containing fluroxalam (active ingredient—N-(2,6-difluorophenyl)-8-fluoro-5-methoxy-[1,2,4]triazolo [1,5-c]pyrimidine-2-sulfonamide). Strain 5(3) was deposited in the collection of microorganisms of the Ufa Institute of Biology (IB UFSC RAS) as UIB-251.

### 2.2. Chemicals and Reagents

The perfluorocarboxylic acids (PFCAs) in the following list were used: perfluorodecanoic acid (PFDA), perfluorononic acid (PFNA), perfluorooctanoic acid (PFOA), perfluoroheptanoic acid (PFHpA) and perfluorooctanesulfonic acid (PFOS) (all high purity > 98%). These were acquired from Sigma Aldrich (St. Louis, MO, USA). Acetonitrile (quality for HPLC) was purchased from Merck (Darmstadt, Germany). All remaining reagents were of analytical-grade grade.

### 2.3. Genome Sequencing, Assembly and Annotation

The isolation of genomic DNA was conducted using a fresh culture biomass (a colony) of strain 5(3), grown on LB agar using the phenol-chloroform method according to the protocol [38]. Sequencing was performed on a GenoLab M system (GeneMind Biosciences Co., Ltd., Luohu, China), and the library was prepared with a ShotGun SG GM library preparation kit (cat. No SG_GM-96, Raissol, Russia). A quality control of the reading was performed using “HTQC” [38]. Low-quality (Q < 25) and short (<100 bp) readings and adapter sequences were removed using Trimomatic v.0.39 software [39]. Raw filtered readings were de novo assembled using SPAdes version 3.15.4 software [40]. We employed Pilon version 1.23 [41] and Bowtie2 version 2.3.5.1 [42] for error correction purposes. In order to validate the circular nature of our assembled replicon, we assessed the presence of overlapping ends. To determine the strain’s species identity, we utilized the average nucleotide identity (ANI) value (https://www.ezbiocloud.net/tools/ani (accessed on 23 October 2023) [43]) and digital DNA-DNA hybridization (DDH) analysis (https://ggdc.dsmz.de/ggdc.php (accessed on 26 October 2023) [44]), with default settings.

We leveraged Prokka software v1.14.5 to annotate the genome [45]. To unravel the functional potential of certain hypothetical proteins, we employed the Basic Local Alignment Search Tool (BLAST) [46].

This genome project has been deposited at GenBank under the accession numbers JAUHUJ000000000 and under BioSample number SAMN36271085, BioProject number PRJNA990579, and SRA accession number SRR25905155.

We used FASTME 2.1.6.1 software [47] to generate a minimum evolutionary tree using intergenomic distances. The tree was rooted at the midpoint [48] and visualized through the online platform https://itol.embl.de/ (accepted date, 30 October 2023) [49]. The clustering of species and sub-species was performed following the protocols described in references [50,51], respectively. To determine the average nucleotide genome identity (ANI) between strain 5(3) and its related strains, we employed the OrthoANI algorithm [52].

### 2.4. Growing Environments and Conditions

Strain 5(3) was grown at 28 °C on Raymond liquid mineral medium (gram per liter of distilled water): NH_4_NO_3_—2.0; MgSO_4_ × 7H_2_O—0.2; KH_2_PO_4_—2.0; Na_2_HPO_4_—3; CaCl_2_ × 6H_2_O—0.01; and Na_2_CO_3_—0.1 [53]. Additionally, PFDA, PFNA, PFOA, PFHpA or PFOS (250 mg/L) was the sole carbon and energy source in an orbital shaker–incubator ES-20/60 (SIA BIOSAN, Riga, Latvia) at 180 rpm.

To obtain the inoculate, the strain was grown on a Raymond mineral medium with the addition of peptone (1 g/L) for 24 h. Prior to inoculation into experimental flasks, the biomass underwent precipitation and washing with sterile distilled water and was added to experimental flasks until 0.1 OD_600_ was reached.

A lysogeny broth (LB) [54], consisting of (per liter of distilled water) 10 g of trypton, 5 g of yeast extract, 5 g of NaCl and 15 g of agar (PanReac, Barcelona, Spain), was used to isolate colonies and determine the bacterial count.

In order to evaluate the efficiency of PFCA degradation, the strain 5(3) was cultivated at a temperature of 28 °C in a liquid mineral medium containing individual C_7_-C_10_ PFCA compounds for 7 days. The experimental protocols were carried out in three autonomous biological repeats.

### 2.5. Isolation and Identification of PFCA Biotransformation Products

The content of PFCA in the medium was evaluated, and the products of their biotransformation were identified on a liquid tandem chromatography-mass spectrometer LCMS-IT-TOF (Shimadzu, Kyoto, Japan) (at the AGIDEL UFSC RAS Equipment Collective Use Center) in ultrafiltrates (≤3 kDa) of culture liquids obtained by ultrafiltration on Vivaflow 50 (Sartorius AG, Göttingen, Germany), as described [21].

The Shim-pack XR-ODS column (75 mm × 2.0 mm id, 2.2 μm) (Shimadzu, Kyoto, Japan) was used for UFLC analysis, in a gradient elution system with 5 mM ammonium acetate in acetonitrile (A) and 0.1% acetic acid (*v*/*v*) in water (B). A linear gradient elution was employed, with the following specifications: from 0 min to 10 min, the composition transitioned from 60% to 30% (B); then, from 10 min to 20 min, the composition reversed, transitioning from 30% back to 60% (B). The chromatographic separation was performed maintaining a steady flow rate of 0.25 mL/min, employing an injection volume of 5 mL. Mass spectrometry data were acquired from an electrospray ionization (ESI) source, which was operated in negative ionization mode. MS operating conditions were as follows: probe voltage, −3.5 kV; curve desolvation line (CDL) and heat block temperature, 200 °C; detector voltage, 1.57 kV; nebulizing gas (N2) flow, 1.5  L/min; collision gas and cooling gas (argon), 50 kPa and 105 kPa, respectively. Parent ion *m*/*z* was acquired in the 150–700 range, with an ion accumulation time of 100  ms (event time, 432  ms; repeat = 3). The timing and separation characteristics of the chromatogram are shown in Appendix A.

The structure of the obtained compounds was determined based on a set of data from the analysis of mass spectra based on the decay of a molecular ion and comparison with literature data.

Metabolic pathways were drawn using ChemSketch 2023.1.1.

The fluoride balance was determined by ion chromatography using an LC-20 Prominence HPLC system with a CDD-10Avp conductometric detector (Shimadzu, Japan). Fluorine ions were separated on a Shodex column (Shodex, New York, NY, USA) at a flow rate of 1 mL/min. An aqueous solution of sodium carbonate and sodium bicarbonate was used as an eluent: 1.8 mM Na_2_CO_3_ + 1.7 mM NaHCO_3_. Post-column eluent suppression was achieved by employing a Xenoic^®^ XAMS suppressor coupled with an ASUREX-A100 (Diduco AB, Umeå, Sweden).

### 2.6. Statistical Analysis

The statistical analysis was conducted utilizing Microsoft Office Excel 2021 software.

## 3. Results and Discussion

### 3.1. Identification and Functional Annotation of the Genome of Strain 5(3)

Cells of the strain 5(3) are Gram-negative, motile, with a single polar flagellum, asporogenous and rod-shaped, able to produce a fluorescent pigment during cultivation on a King B medium. On nutrient agar, they form circular and non-pigmented colonies. The growth temperature range lies in interval of 10–36 °C; the optimal growth temperature is between 26 and 30 °C. The optimal pH value is 6.8–8.0. They do not reduce nitrate to nitrite. Arginine dihydrolase, catalase and cytochrome oxidase are produced. The Voges–Proskauer reaction is negative.

The strain did not hydrolyze lecithin or starch but hydrolyzed gelatin. The strain did not synthesize lipolytic enzymes and was not able to grow on a medium with twin-80. At a concentration of 0–5% NaCl, intensive growth was observed, and at a higher concentration of up to 10% NaCl, weak growth was observed. The strain 5(3) did not use xylose, rhamnose, galactose, malonate, D-tartrate, benzoate, or L-tryptophan as the sole source of carbon and energy. Ribose, glucose, D-mannose, D-mannitol, ketogluconate, L-histidine, L-rhamnose, histamine and D-sorbitol were used as carbon sources.

Based on a preliminary analysis of physiological and biochemical data and the 16S rRNA gene, the strain was identified as belonging to the *Pseudomonas* genus. Pseudomonads have a large number of assemblies represented in various databases such as NCBI, BV-BRS, TYG, and Pseudomonas.com [55]. Detailed taxonomy of this genus is still lacking and despite a long history of study and extensive databases, it is constantly being supplemented with new findings. As the result of the analysis of genome-wide sequencing, the strain 5(3) (Figure 1A) was reliably referred to as Pseudomonas mosselii, and the values of the main identification parameters relative to the reference (*Pseudomonas mosselii* DSM17497T) are as follows: ANI, 97.40%, and DDH, 80.30% (with species thresholds > 95% and >70%, respectively [56]).

The sequencing and complete genome assembly of the strain *Pseudomonas mosselii* 5(3) revealed that the assembly contained 5134 coding sequences (CDSs), four rRNA clusters, and 65 tRNAs. The genome consists of 73 contigs with a total length of 5,676,241 bp. The N50 value is 165,927 bp, and the GC content is 64.38%.

Of the 5134 genes, 2895 (56.8%) were functionally annotated (Figure 1B).

The result of the functional annotation shows that strain 5(3) possesses genes of all metabolic pathways necessary for the existence of an autonomous culture.

The category of “Xenobiotics biodegradation and metabolism” in strain 5(3) contains more than 60 genes. Most of this category is associated with the degradation of various aromatic compounds (including ethylbenzene, nitrotoluene xylene, styrene, naphthalene, polycyclic aromatic hydrocarbon, aminobenzoate, dioxin and atrazine). According to [57], laccases and dehalogenases emerge as the most promising contenders for PFCA biotransformation enzymes. These enzymes are currently under the scrutiny of scientists investigating biocatalysis as a tool for removing persistent pollutants, including PFCA. Indeed, the putative enzymes potentially involved in PFCA decarboxylation and decarbonylation were found within the investigated genome with a 50% occurrence rate. The genes that could participate in the destruction of fluorinated organic compounds (dehalogenation) include the haloalkane dehalogenase gene (dhaA) and the haloacetate dehalogenase H-1 gene (dehH1). Like the previously characterized fluoracetate dehalogenases [58], they can mediate the defluorination of PFCA at the alpha-carbon position.

And the mechanisms of resistance to them can be mediated by the presence of the following:-Decarboxylase gene (novR), which can carry out several successive steps of the oxidative decarboxylation of PFCA; this was previously described and shown, using a chemical method, for similar compounds [59];-Alkanesulfonate monooxygenase gene (ssuE), catalyzing the desulfonation of various organosulfonate substrates under conditions of sulfate starvation, which can also produce a protective role under conditions of oxidative stress [60];-Fluoride ion transporter gene (crcB). With an increase in fluoride levels, it enhances the transcriptional activity of genes located further downstream. It is assumes that these genes aid in reducing the detrimental impacts caused by excessive concentrations of fluoride [61], and assumes that many genes are regulated by them. According to [62], effective defluorination activity will require a high level of tolerance to elevated intracellular concentrations of F^−^ in the microorganism.

Pseudomonas genomes possess a diverse array of enzymes with the potential to facilitate defluorination reactions, utilizing both oxygenase-mediated biodegradation processes and anaerobic metabolic activities. While the complete functionality of these enzymes in defluorination mechanisms remains undisclosed, it is conceivable that these degradative enzymes might contribute to Pseudomonas’ response to intracellular fluoride release.

In none of the previous publications devoted to the destruction of organofluorine compounds by microorganisms have the authors found any mention of the confirmed presence of the presented complex of enzymes.

It should be remembered that Pseudomonas genomes encode > 25% of proteins with unknown function, which suggests that the new defluorinating activity may belong to them [63,64].

### 3.2. Growth on C_7_-C_10_ PFCA and Their Defluorination

To grow on PFCA, bacteria need to transport it into the cell, develop an efficient enzyme capable of catalyzing the cleavage of the strong C-F bond, and detect a toxic fluorine ion and remove it from the cell. And it seems that this was not a problem for the *P. mosselii* strain 5(3). The strain actively grew on mineral media containing C_7_-C_10_ PFCA as the only source of carbon (Figure 2A–E), reaching the highest point of optical density of the culture liquid after 6–7 days, and the maximum optical density was in the range of 0.7–1.0 depending on the acid and was directly related to the quantity of carbon atoms of the PFCA. With the growth of the strain on C_7_-C_10_ perfluorocarboxylic acids, their complete decomposition took place after 7 days, but 5 days were enough for the bacteria to transform PFOS.

The results obtained in this study are significantly superior to those presented earlier. The closest result to us was declared by the authors in work [23] and was achieved using a strain of *Acidimicrobium* sp. A6 for the removal of PFOA and PFOS in an aerobic environment, where the degree of biodegradation did not exceed 63% for a concentration of 100 mg/L for 100 days.

Only in contrast to them, we observed the release of F^−^, which is an indicator of partial decomposition and perhaps these ions cause the inhibition of the further destruction of intermediate fluorinated compounds. The transformation of perfluorinated substrates was in parallel with the release of free F^−^ into the medium, and the beginning of release correlated with the beginning of a linear decrease in their concentration in the medium. The absence of an inhibitory effect on bacterial growth is precisely explained by the presence of the fluoride ion transporter gene, which allows the bacteria to cross the physiological barrier of fluoride toxicity during the decomposition of PFCAs, and to carry out defluorination, regardless of the difference in redox potentials of the processes, at the level of dechlorination of the polychlorinated compounds.

### 3.3. Biodegradation of C_7_-C_10_ PFCAs

The biological degradation of PFCAs may include several steps, including decarboxylation, hydroxylation and defluorination reactions.

During chromatographic analysis with mass spectrometry in the initial culture liquid when cultivating strain 5(3) in a medium with PFHpA, we observed its dissociated acid ions (molecular ion with an *m*/*z* ratio of 363) (Figure 2F). A day later, a compound with *m*/*z* 319 was detected in the ultrafiltrate, which may occur as a result of the elimination of the COO^−^ group (*m*/*z* 44) from the carboxyl group. Its concentration continued to increase up to 4 days, after which a component with *m*/*z* 313, identified as perfluorohexanoic acid (PFHxA), began to appear in the medium, presumably under the action of the enzyme haloalkane dehalogenase. After 6 days, this compound prevailed in the culture liquid. A similar series of reactions took place during the conversion of homologous acids, including PFOA (*m*/*z* 413) (Figure 2G), PFNA (*m*/*z* 463) (Figure 2H), and PFDA (*m*/*z* 513) (Figure 2I). Intermediate products between cycles differed by the amount of mass of the cleaved CF_2_^−^ group, i.e., at *m*/*z* 50. Everywhere, PFHxA was found as the final prevailing destruction product in an amount from 50 to 95% of the total destruction products. It took two such reaction cycles to decompose PFOA, three cycles to decompose PFNA, and four cycles to decompose PFDA.

PFCA compounds, which also include C−S and C−C bonds, have the potential for one-electron reduction or oxidation, followed by hydroxylation.

Similarly, we assumed that sulfonate group in PFOS would first break with the release of sulfite for bacterial metabolism, and that the synthesis of perfluorooctane followed by the release of hydroxyl and carboxyl groups and defluorination reactions results in the formation of perfluoroheptanoic acid, as shown in our previous work [21]. But in the realities of the experiment, everything was different. During the decomposition of PFOS (*m*/*z* 499) (Figure 2J), probably with the participation of haloacetate dehalogenase H-1, free electrons attacked the most vulnerable spot, the α-position of the fluorine of the alkyl chain, replacing two fluorine atoms with hydrogen, followed by the elimination of the carbon–carbon bond of the alkyl chain through an intermediate unidentified a product with *m*/*z* 461, i.e., shortening the chain by CH_2_ with its release and the formation of perfluoroheptane sulfonate (*m*/*z* 449) by day 2 of cultivation. This was followed by a similar cycle and the formation of perfluorohexane sulfonate (*m*/*z* 399) by day 4. And only after that, under the probable action of alkanesulfonate monooxygenase, the C−S bond was broken, and SO^−^_3_ was released to form a compound with *m*/*z* 319, with its transformation into PFHxA, and after 7 days, this compound was detected in an amount of at least 70% of the sum of the destruction products.

Thus, in the final stage of cultivation, the compounds detected in the medium dissociated acid ion led to the identification of perfluorohexanoic acid. The obtained data allowed us to construct an assumed the scheme of destruction of C_7_-C_10_ PFCA (Figure 3). Further investigation of the decomposition of substances by the studied strain is currently not possible and requires further study of its catabolic capabilities. From available sources, lower-molecular-weight fragments produced by PFCA destruction were observed only in [23] but without confirming the described genetic potential and partial mineralization by fluorine release. The literature on the toxicity of PFCA indicates that the shorter the F−C chain, the more favorable its toxicological profile. Accordingly, there is a clear rationale for reducing the toxicity of PFCA in relation to human exposure, even by shortening its chain by one link [65,66,67].

### 3.4. Fluoride Balance

In our case, it could be assumed that the destruction stops as the fluoride ion level in the medium reaches 19 mg/L (1 mM), the value at which, according to [68,69], this anion becomes toxic at intracellular levels. But in our early studies, microorganisms remained viable at concentrations exceeding 150 mg/L [70]. Therefore, the suppression of PFCA defluorination by its biodegradation products is more likely. And for cells initially having the crcB gene, which encodes the fluoride exporting protein [71] and controls the level of fluoride, when a potentially toxic concentration of it occurs, they organize its displacement due to an electrochemical gradient, which is shown by the example of P. putida KT2440 [61].

According to the results obtained and the scheme presented, during biodegradation, 26.3 (growth on PFHpA), 46.0 (on PFOA), 61.6 (on PFNA), 74.1 (on PFDA), and 57.1 (on PFOS) mg/L fluoride ions should have been released into the culture medium according to the calculated data. In contrast to the expected value, the actually detected concentration in all the studied variants did not exceed 24.4 mg/L, i.e., the degree of extraction of F- only in the case of growth on PFHpA was stoichiometric and tended to 100%, but in other variants, this value did not exceed 48.5%. Some authors explain the non-stoichiometric release of fluorine during biodegradation by its redirection by the cell for their needs. For example, Pseudomonas sp. strain 273 is a destructor of bichlorinated and fluorinated alkanes with medium and long chain (C_7_-C_16_)—for phospholipid biosynthesis [72,73,74].

## 4. Conclusions

The *Pseudomonas mosselii* strain 5(3), which was isolated from herbicide-contaminated soil, capable of destruction of C_7_-C_10_ PFCA, was described for the first time. The genome of strain 5(3) is fully sequenced, contains a total of 5134 genes—in particular, genes not mentioned in microorganisms before—including the haloalkane dehalogenase gene (*dhaA*), haloacetate dehalogenase H-1 gene (*dehH1*), fluoride ion transporter gene (*crcB*) and alkanesulfonate monooxygenase gene (*ssuE*), which are crucial for the degradation of fluorinated compounds. The strain of *P. mosselii* 5(3), after 7 days of batch-cultivation with diverse of C_7_-C_10_ PFCAs as the sole resource of carbon and energy, completely utilized them. The results of the LC-MS analysis showed that the transformation takes place due to perfluorohexanoic acid with the release of various levels of stoichiometry (depending on PFCAs) of mineralization indicators—fluorine ions determined by ion chromatography. Thus, the *Pseudomonas mosselii* strain 5(3) demonstrates a genetically confirmed high potential for the decomposition of C_7_-C_10_ PFCAs, and the results obtained will help both in the fundamental exploration of cellular processes of the strain and in the practice of developing biotechnologies for environmental rehabilitation from contamination with organofluorine compounds.

## Figures and Tables

**Figure 1 toxics-11-01001-f001:**
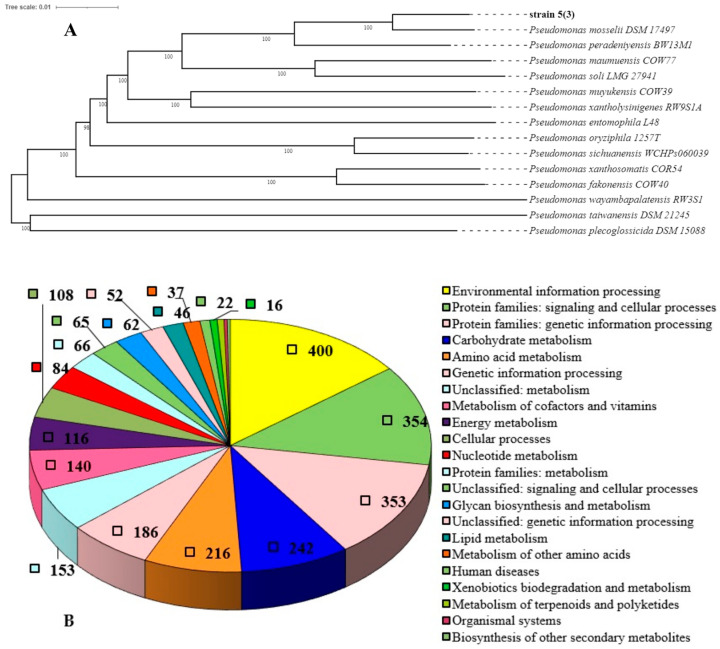
Whole-genome tree of the position of strain 5(3) within the genus Pseudomonas (**A**) and the number of genes associated with common functional categories in its genome according to the KEGG classification (**B**).

**Figure 2 toxics-11-01001-f002:**
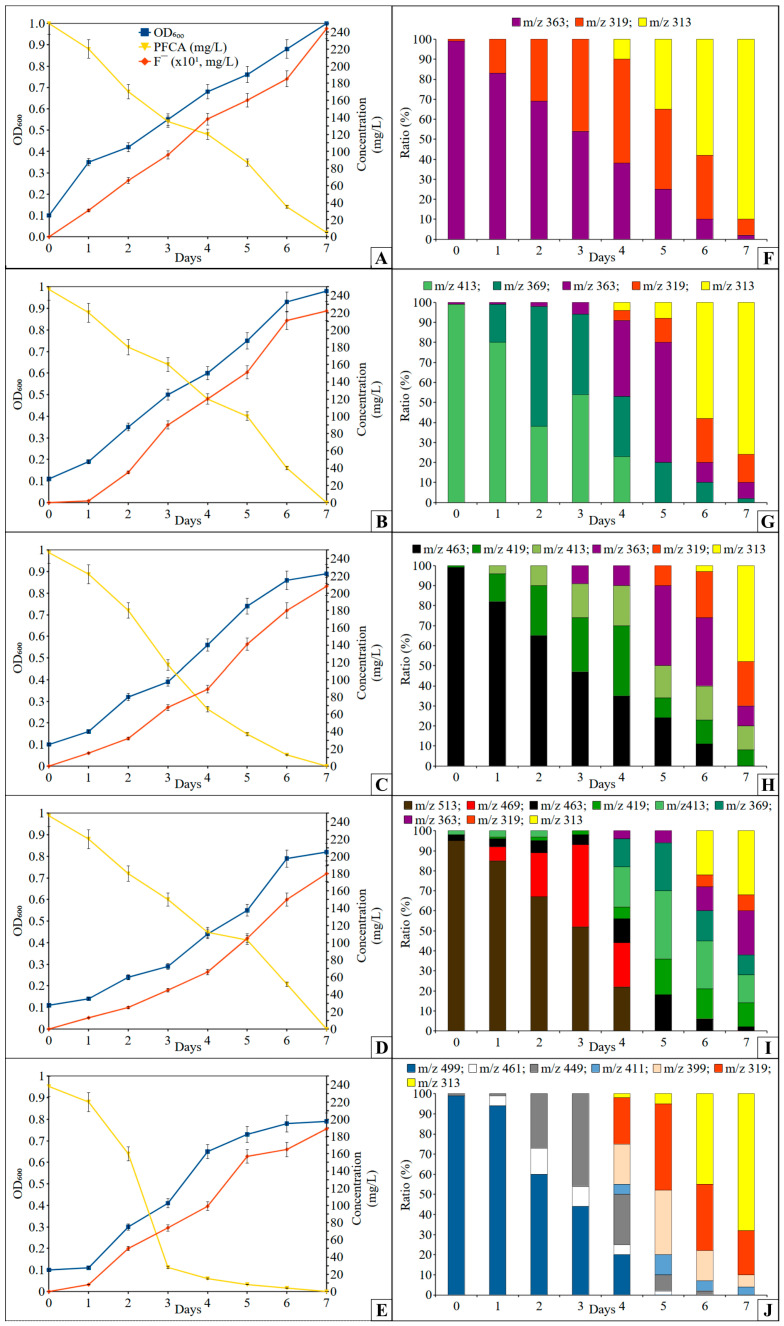
Dynamics of growth of *Pseudomonas mosselii* strain 5(3) (OD_600_), release of fluorine ions, changes in PFCA concentration (graphs on the left) and percentages of its and its degradation products (according to *m*/*z*) (histograms on the right) during cultivation in a liquid mineral medium: (**A**,**F**)—PFHpA; (**B**,**G**)—PFOA; (**C**,**H**)—PFNA; (**D**,**I**)—PFDA; (**E**,**J**)—PFOS.

**Figure 3 toxics-11-01001-f003:**
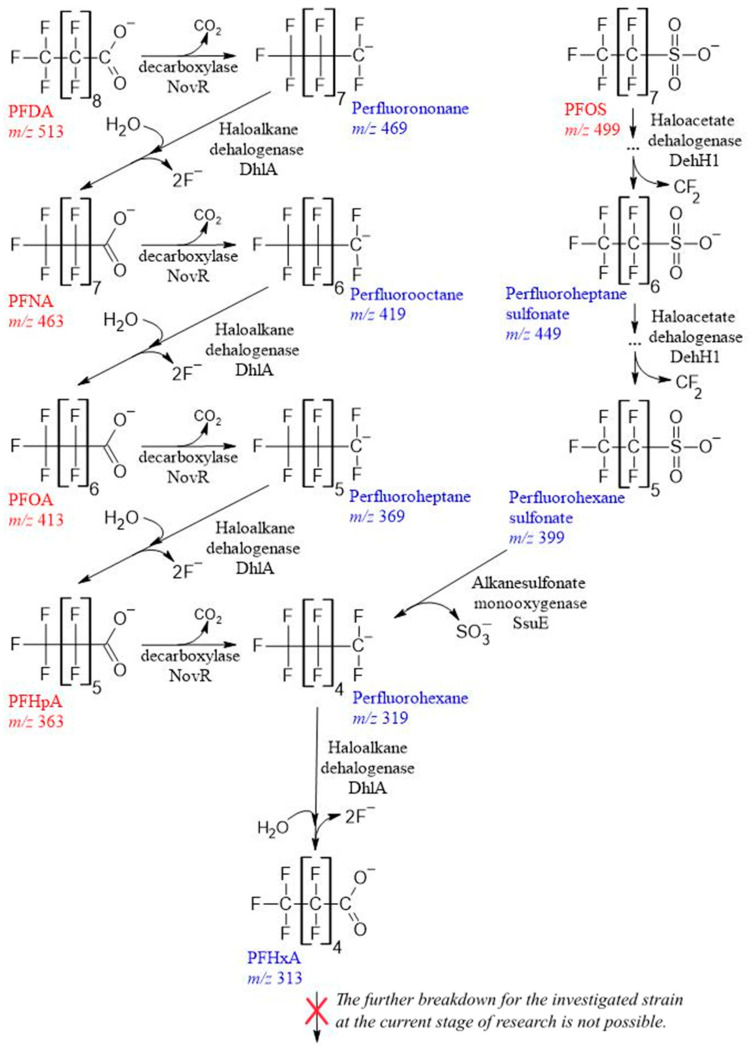
The proposed C_7_-C_10_ PFCA biodegradation pathways of *Pseudomonas mosselii* 5(3) on the basis of metabolite analysis by LC-MS, ion chromatography and the results of whole genome sequencing.

## Data Availability

The data that support the findings of this study are available from the corresponding author on reasonable request.

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
