# Peer review of "Biodegradation Potential of C7-C10 Perfluorocarboxylic Acids and Data from the Genome of a New Strain of Pseudomonas mosselii 5(3)"

_toxics, 2023, doi:10.3390/toxics11121001_

Round 1

Reviewer 1 Report

Comments and Suggestions for Authors

The manuscript describes the isolation and potential for C7-C10 perfluoro-carboxylic acids degradation of a new Pseudomonas strain. The study has some strong points: it is the first genomic characterization of Pseudomonas mosselii 5(3); the results of PFCA degradations and annotation are coherent and well discussed. I recommend the publication of this manuscript with minor corrections, as stated below.

146-147 - The unit “gram” is missing.

182 – I didn’t find the supplementary material.

206 – Replace “he did not synthesize lipolytic enzymes” with “the strain did not synthesize lipolytic enzymes

Please revise the punctuation; the use of commas and dots needs a careful look.

244-249 - Confused.

272, 350 – Pseudomonas in italic.

357-358 – The sentence appears to be incomplete. You did mean that according to [67], suppression of PFCA defluorination by their biodegradation products is probable.

359-362 – That sentence belongs to the previous paragraph. 

Comments on the Quality of English Language

The English is clear and easy to read but needs revision regarding punctuation.

Author Response

Thank you for your thorough review of our article. We have made the necessary corrections according to your recommendations. We hope that the revised version of the article will meet your expectations.

146-147 - The unit “gram” is missing.

Thank you very much. We have made corrections.

182 – I didn’t find the supplementary material.

We apologize for the inconvenience. However, we have uploaded all additional materials of the manuscript on the website. There might have been a technical glitch.

206 – Replace “he did not synthesize lipolytic enzymes” with “the strain did not synthesize lipolytic enzymes

Corrected. Thank you very much.

Please revise the punctuation; the use of commas and dots needs a careful look.

244-249 – Confused.

Thank you for pointing out the unclear parts in the manuscript. We have made corrections to the text to make it more readable and understandable.

272, 350 – Pseudomonas in italic.

Corrected. Thank you very much.

357-358 – The sentence appears to be incomplete. You did mean that according to [67], suppression of PFCA defluorination by their biodegradation products is probable.

Thanks for the comment. We have made changes to make it clear to the reader.

 359-362 – That sentence belongs to the previous paragraph. 

Corrected. Thank you very much.

The English is clear and easy to read but needs revision regarding punctuation.

We have reviewed the article and made corrections

We would like to express our gratitude to you for such a thorough and precise review. We have made all the necessary corrections. We sincerely hope that the work, in its current form, will satisfy you.

Reviewer 2 Report

Comments and Suggestions for Authors

The experiment was well designed and presented. It seems almost unbelievable that 200 mg/L of PFOA can be completely degraded in 7 days. Have the authors try the degradation of a mixture of PFOA compounds simultaneously? How does the treatment change the toxicity of the solution?

Comments on the Quality of English Language

English needs to be improved significantly.

Author Response

The experiment was well designed and presented. It seems almost unbelievable that 200 mg/L of PFOA can be completely degraded in 7 days. Have the authors try the degradation of a mixture of PFOA compounds simultaneously? How does the treatment change the toxicity of the solution?

Comments on the Quality of English Language

English needs to be improved significantly.

                The authors have been engaged in isolating microorganisms capable of breaking down and utilizing perfluorinated organic compounds for a long time. There are no discoveries without luck. And to isolate such a strain of bacteria was lucky and fortunate for us.

The work of our team on the problem of destruction of perfluorinated organic compounds has been carried out within specific projects (in this case, the work was carried out in accordance with a grant from the Russian Science Foundation), where specific tasks and their timelines are given. The work on decomposing PFCA mixtures is currently not included in these projects and has not been carried out. The presented work focuses on individual compounds and serves as the development of methods for determination, analysis, identification of intermediate components, etc., and it is only one of the subsequent steps that involves working with PFCA mixtures.

Data on the comparative toxicity of short-chain (C6) and long-chain (C8 and above) PFCA has been comprehensively systematized and examined for the first time in the study (Luz A.L., Anderson J.K., Goodrum P., Durda J. Perfluorohexanoic acid toxicity, part I: Development of a chronic human health toxicity value for use in risk assessment. Regul Toxicol Pharmacol. 2019; 103:41-55. doi:10.1016/j.yrtph.2019.01.019) Data on the comparative toxicity of short-chain (C6) and long-chain (C8 and above) perfluorocarboxylic acids (PFCAs) has been comprehensively systematized and examined for the first time in this study, with subsequent confirmation (one of the latest reviews on this topic Zango Z.U., Ethiraj B., Al-Mubaddel F.S., Alam M.M., Lawal M.A., Kadir H.A., Khoo K.S., Garba Z.N., Usman F., Zango M.U., Lim J.W. An overview on human exposure, toxicity, solid-phase microextraction and adsorptive removal of perfluoroalkyl carboxylic acids (PFCAs) from water matrices. Environ Res. 2023; 231(Pt 2): 116102. doi:10.1016/j.envres.2023.116102). All observed effects associated with PFHxA were mild and/or reversible and were noted at levels significantly higher than with PFOA. PFHxA has a faster elimination kinetics from the human body compared to PFOA (t½ ~32 days versus 3.5 years) and is not considered as bioaccumulative. For PFHxA, the Provisional Oral Reference Dose (RfD) for oral exposure is calculated to be 0.25 mg/kg/day, which is four orders of magnitude higher than the RfD for chronic oral exposure calculated by the US Environmental Protection Agency for PFOA (0.00002 mg/kg/day). In another study, it was shown that short-chain PFCA have a lesser impact on the skin and organs compared to PFOA (Han J.S., Jang S., Son H.Y., Kim Y.B., Kim Y., Noh J.H., Kim M.J., Lee B.S. Subacute dermal toxicity of perfluoroalkyl carboxylic acids: comparison with different carbon-chain lengths in human skin equivalents and systemic effects of perfluoroheptanoic acid in Sprague Dawley rats. Arch Toxicol. 2020 Feb;94(2):523-539. doi: 10.1007/s00204-019-02634-z). Thus, the literature on the toxicity of PFCA indicates that the shorter the F-C chain length, the more favorable its toxicological profile. Accordingly, there is a clear rationale for reducing the toxicity of PFCA in relation to human exposure even by shortening the chain length by one link.

            The revised article has been modified in terms of English language after consultation with our colleague, Professor Kudoyarova G.R., a regular author of articles in MDPI journals, as well as a scientific researcher whose native language is English.

Reviewer 3 Report

Comments and Suggestions for Authors

The manuscript investigates the ability of Pseudomonas mosselii 5(3) to biodegrade PFCAs and is overall interesting and well-structured. Please check some editing mistakes and the use of commas and full stops in the text (lines 13-14, line 42, line 45, lines 206-207). For lines 269-271, please rephrase, line 272 please use italics. The authors could briefly reference other Pseudomonas species that have been applied in similar bioremediation processes and the type of pesticides present in contaminated soil samples. The physicochemical characteristics of the soil could be helpful and linked with the performance of Pseudomonas mosselii 5(3). The concentration of pesticides should be included.  

Comments on the Quality of English Language

Please check Comments and Suggestions for the authors

Author Response

The manuscript investigates the ability of Pseudomonas mosselii 5(3) to biodegrade PFCAs and is overall interesting and well-structured. Please check some editing mistakes and the use of commas and full stops in the text (lines 13-14, line 42, line 45, lines 206-207). For lines 269-271, please rephrase, line 272 please use italics. The authors could briefly reference other Pseudomonas species that have been applied in similar bioremediation processes and the type of pesticides present in contaminated soil samples. The physicochemical characteristics of the soil could be helpful and linked with the performance of Pseudomonas mosselii 5(3). The concentration of pesticides should be included.  

The authors express their deep gratitude for the tremendous effort that has been made. All errors have been corrected. Lines 13-14, line 42, line 45, lines 206-207 were revised and corrected. Lines 269-271 were rephrased. Line 272 was corrected.

            We want to express our gratitude to the reviewer for reminding us to provide data on known strains of Pseudomonas bacteria that are capable of degrading PFCA. The list of such strains is not extensive, as only a few strains have been identified that can transform perfluorocarboxylic acids. These strains are specifically associated with the biodegradation of PFOA and PFOS, and we could not find any mentions of their involvement with other PFCA in the literature. In the context of this article, P. parafulva YAB-1 is mentioned (which can degrade 32.2% of perfluorooctanoic acid at an initial concentration of 500 mg/L) as a degrader with identified putative haloacid dehalogenases and fluoroacetate dehalogenase, which may be involved in the degradation of fluorinated compounds. Additionally, data on P. aeruginosa HJ4 has been added (which can degrade PFOS by 67% at a concentration not exceeding 2 mg/L). The authors agree with the reviewer on the importance and usefulness of knowing the physicochemical characteristics of the soil from which the studied bacterial strain was isolated. We have this data available as these are mapped soils of agricultural land. We provide this information in the text of the article. We also know about the composition of pesticide contamination (unfortunately, we only have qualitative data, not quantitative): a mixture of herbicides including phosphorus-containing glyphosate (active ingredient (a.i.) - N-(phosphonomethyl)glycine), chlorine-containing 2,4-D (a.i. - 2-(2,4-dichlorophenoxy)acetic acid), and fluorine-containing florasulam (a.i. - N-(2,6-difluorophenyl)-8-fluoro-5-methoxy-[1,2,4]triazolo[1,5-c]pyrimidine-2-sulfonamide).

Round 2

Reviewer 2 Report

Comments and Suggestions for Authors

My previous comments have been addressed.

Comments on the Quality of English Language

NA